# Endothelial cells protect *Schistosoma mansoni* from hydrogen peroxide-induced death

**Bruna Oliveira Lopes Souza[1], Ronald Alves dos Santos[1], Kelvin Edson Marques de Jesus[1], Juliana Bezerra Dória Lima[1], Andressa Moreira Lima[1], Brenda Rodrigues Brito Cunha Silva[1], Fernanda Freitas Costa[1], Lorena Conceição de Queiroz[1], Camilla Almeida Menezes[1], Sânzio Silva Santana[1], Karine Araujo Damasceno[1], Isadora Cristina de Siqueira[1], Marilda de Souza Gonçalves[1], Dalila Luciola Zanette[2], Thassila Nogueira Pitanga[1], Ricardo Riccio Oliveira📷[1]***

**1** Global Health and Neglected Diseases Research Laboratory, Gonçalo Moniz Institute, Oswaldo Cruz Foundation (Fiocruz), Salvador, Bahia, Brazil, **2** Laboratory for Applied Science and Technology in Health, Carlos Chagas Institute, Oswaldo Cruz Foundation (Fiocruz), Curitiba, Paraná, Brazil

* ricardo.riccio@fiocruz.br

## Abstract

### Introduction

*Schistosoma mansoni*, the causative agent of intestinal schistosomiasis, thrives in the human host, particularly within the vascular system. Understanding the role of endothelial cells during infection is crucial. Currently, schistosomiasis treatment depends solely on praziquantel (PZQ), but emerging evidence suggests decreasing efficacy. This highlights the need for new therapeutic strategies, including agents that modulate the host antioxidant response, such as dapsone.

### Methods

Adult *S. mansoni* worms were harvested from infected mice via portal perfusion. Human umbilical vein endothelial cells (HUVECs) were cultured and exposed to worm pairs and PZQ for 1, 3, or 6 hours. Post-exposure, RNA was extracted and analyzed by qPCR to assess the expression of antioxidant genes (*NRF2, SOD1, GPx, GSR, CAT*). Additionally, worm viability under oxidative stress was evaluated by incubating worms with hydrogen peroxide ($H_2O_2$), in the presence or absence of HUVECs, catalase, or dapsone hydroxylamine.

### Results

Worms did not significantly alter expression of host antioxidant genes except for catalase. $H_2O_2$ exposure led to worm death, but co-incubation with HUVECs improved worm viability and survival, suggesting a protective role of endothelial cells against oxidative stress. Furthermore, dapsone hydroxylamine reversed the protective effect

**Data availability statement:** All raw data supporting the findings of this study are provided in the Supporting Information files published alongside the article.

**Funding:** This work was supported by the "Programa de Excelência em Pesquisa" (PROEP/IGM/2020, Edital 01/2020, Grant ID: FIOTEC IGM-002-FIO-20-2-24, to RRO); "Coordenação de Aperfeiçoamento de Pessoal de Nível Superior" (CAPES) – Finance Code 001 (doctoral fellowship to BOLS); and "Fundação de Amparo à Pesquisa do Estado da Bahia" (FAPESB, master's fellowship to BOLS). The funders had no role in study design, data collection and analysis, decision to publish, or preparation of the manuscript.

**Competing interests:** The authors have declared that no competing interests exist.

of catalase, reducing worm viability. However, worms remained viable in co-culture with HUVECs, indicating additional, unidentified mechanisms of protection.

## Conclusion

Endothelial cells may play a key role in protecting *S. mansoni* against host oxidative defenses. Dapsone hydroxylamine interferes with this protection by inhibiting catalase activity. These findings point to potential therapeutic strategies targeting the host-parasite interface and the antioxidant environment in schistosomiasis.

### Author summary

*Schistosoma mansoni*, the causative agent of the intestinal form of the disease, is well adapted to its human host. Unlike other intestinal helminths, the adult *S. mansoni* worm has a peculiar habitat: the vascular lumen. Despite this known fact, there is still limited information about the interaction between the adult worm and endothelial cells. Current treatment relies exclusively on praziquantel (PZQ), which has shown signs of decreasing efficacy. This underscores the urgent need for alternative therapies or drug combinations, including those that target the host's antioxidant response - such as dapsone. We evaluated the interaction between adult worms and HUVECs, and their ability to protect the parasites from oxidative stress induced by hydrogen peroxide ($H_2O_2$). It was observed that the worms did not significantly alter the expression of antioxidant genes by HUVECs, except for catalase. $H_2O_2$ was lethal to the worms, but this effect was reversed in the presence of the endothelial cells, indicating a protective effect. We also tested dapsone hydroxylamine, which inhibited catalase activity and reduced worm viability. However, when the worms were cultured with endothelial cells, they remained viable, suggesting other protective mechanisms. These findings contribute to new therapeutic approaches against schistosomiasis.

## Introduction

Schistosomiasis is a neglected tropical disease (NTD) that disproportionately impacts populations living in poverty, particularly those lacking access to safe water and basic sanitation services. According to the World Health Organization, schistosomiasis affects an estimated 250 million individuals worldwide, places approximately 779 million at risk of infection, and remains endemic in 78 countries - primarily in Africa, but also in parts of Asia and South America [1]. *Schistosoma mansoni* is one of the main species responsible for intestinal schistosomiasis [2], and the parasite is highly adapted to the human host.

Given the intravascular habitat of adult *S. mansoni* worms, understanding the role of endothelial cells during infection is of fundamental importance. Although some investigations have addressed parasite-host interactions involving endothelial

cells, investigations specifically focused on the interface between adult worms and the endothelium remain scarce. Most existing research has examined the interaction between endothelial cells and *S. mansoni* eggs [3]. These studies have demonstrated that eggs can bind to platelets or host plasma proteins such as von Willebrand factor, potentially facilitating their adhesion to the endothelium and promoting endothelial cell activation [4,5]. Moreover, eggs and egg antigens have been shown to induce proliferation and apoptosis of endothelial cells in vitro [6,7]. Endothelial interaction with the egg also contributes to granuloma formation [8], promoting angiogenesis during hepatic granuloma development [7,9,10]. These findings highlight the active participation of endothelial cells in the host response to *S. mansoni* eggs. However, much less is known about how adult worms interact with endothelial cells. In this context, investigating whether adult worm-endothelium interactions modulate oxidative stress pathways is of particular interest. Notably, the expression of antioxidant genes by endothelial cells during schistosomiasis has not yet been evaluated.

Although the immune system plays a critical role in the defense against the parasite, there are no reports of spontaneous cure in untreated individuals. Currently, treatment for schistosomiasis relies solely on praziquantel (PZQ), which, despite its considerable efficacy, has been associated with reduced cure rates and therapeutic failure in some studies [11,12]. These limitations underscore the urgent need to identify new therapeutic agents or combination strategies. In this context, host-directed therapies that target inflammation or oxidative stress pathways have gained attention. Dapsone, an anti-inflammatory and antioxidant compound, has been investigated in other chronic inflammatory conditions and may serve as a candidate for further evaluation in the context of schistosomiasis [13,14].

In this study, we investigated the interaction between adult *S. mansoni* worms and human umbilical vein endothelial cells (HUVECs), focusing on the ability of these cells to protect the parasite from death under pro-oxidative conditions. In addition, we evaluated the in vitro effect of dapsone hydroxylamine, an active metabolite of dapsone, on parasite viability in an oxidative environment, aiming to explore its potential to induce worm death through catalase inhibition.

## Methods

### Ethics statement

All experiments involving animals were approved by the Ethics Committee on Animal Use of the Gonçalo Moniz Institute – Fiocruz Bahia (protocol 015/2021).

### Recovery of adult worms

Swiss Webster mice were infected subcutaneously with 100 *S. mansoni* cercariae. After 6–8 weeks of infection, mice were euthanized using a lethal dose of ketamine + xylazine administered intraperitoneally. Adult *S. mansoni* worms were recovered through portal perfusion using 0.9% saline solution containing 3% sodium citrate [15,16]. Recovered worms were transferred to a Petri dish containing RPMI 1640 medium (Gibco, Life Technologies Australia Pty Ltd, Mulgrave, VIC, Australia), and then to a second dish containing RPMI supplemented with 10% fetal bovine serum (Gibco, New York, NY, USA), and 1% glutamine and HEPES. Prior to the experiments, worms were placed in a 96-well plate, and their sex and viability were assessed under an inverted microscope.

### Cell cultures for gene expression analysis

Immortalized HUVECs were seeded in flat-bottom 12-well plates at a concentration of $1 \times 10^5$ cells/mL in 800 µL of RPMI 1640 medium supplemented with 1% PenStrep (10,000 U/mL penicillin and 10,000 µg/mL streptomycin; Gibco, New York, NY, USA), 10% fetal bovine serum, and 1% glutamine and HEPES (GH). Cells were maintained in a humidified incubator at 37°C with 5% $CO_2$. After 24 hours of incubation, one pair of adult worms was added per well, and plates were cultured for 1, 3, or 6 hours. All experiments were performed in triplicate and repeated at least three independent times.

## Quantitative real-time PCR (qRT-PCR)

At the end of the incubation period, the worms and supernatant were removed, and Trizol reagent (Thermo Fisher Scientific) was added to each well for mRNA extraction. RNA concentration and purity were measured using a NanoDrop spectrophotometer.

Complementary DNA (cDNA) synthesis was performed using 250 ng of total RNA with the High-Capacity cDNA Reverse Transcription Kit (Applied Biosystems, Foster City, CA, USA) following the manufacturer's instructions. The thermal conditions were 25°C for 10 minutes, 37°C for 120 minutes, 85°C for 5 minutes, and finally 4°C. The resulting cDNA was resuspended in RNase-free water, aliquoted, and stored at –20°C.

Gene expression analysis was performed using real-time PCR (qPCR) in 96-well optical plates, with sample duplicates, on the ABI 7500 Real-Time PCR System (Applied Biosystems, Foster City, CA, USA). Antioxidant genes and the endogenous control gene HPRT (Table 1) were amplified using SYBR-Green PCR Master Mix (Applied Biosystems, Foster City, CA, USA), according to the manufacturer's protocol.

The standard qPCR conditions were as follows: initial denaturation at 95°C for 10 minutes, followed by 40 cycles of 95°C for 15 seconds and 60°C for 60 seconds. After amplification and melt curve analysis, threshold cycle (Ct) values were obtained using the 7500 system software (Applied Biosystems, USA). Gene expression levels were normalized to the endogenous control (HPRT), and relative expression values (ddCt) were calculated using the median of untreated controls as the calibrator.

## Evaluation of *Schistosoma mansoni* adult worm viability under oxidative conditions

Three experimental models were used to assess the survival of adult *S. mansoni* worms under oxidative stress. In the first model, worm pairs were cultured in RPMI medium without HUVECs, supplemented with five different concentrations of hydrogen peroxide ($H_2O_2$): 100 μM, 200 μM, 400 μM, 800 μM, and 1600 μM. The second model consisted of co-cultures of HUVECs and worm pairs exposed to the same $H_2O_2$ concentrations. In the third model, worm pairs were cultured in RPMI medium supplemented with the same $H_2O_2$ concentrations and the addition of catalase enzyme. All experimental models were incubated at 37°C with 5% $CO_2$ for 24 hours. Experiments were performed in quadruplicate and repeated twice.

Following incubation, worm viability was assessed using an inverted microscope. Viability was evaluated based on motility and tegument integrity, using a scoring system ranging from 0 to 3 (Fig 1) [17]. Score 0 indicated death, defined by complete tegument darkening and absence of movement. Score 1 indicated a darkened tegument with low motility and impaired oral and ventral suckers, often with worms detached from the well surface. Score 2 indicated partial tegument damage and moderate motility. Score 3 indicated fully viable worms with intact tegument and active movement. Egg deposition in wells was also considered a sign of viability. Scoring was performed independently by two observers, and in cases of disagreement, re-evaluation was carried out to reach a consensus.

**Table 1. Primer sequences for qPCR amplification of antioxidant and control genes.**

| Gene | Forward Primer (5'→3') | Reverse Primer (5'→3') |
|------|------------------------|------------------------|
| NRF2 | GTA TGC AAC AGG ACA TTG AGC | ATG GTA GTC TCA ACC AGC TT |
| SOD1 | TGG CCG ATG TGT CTA TTG AA | CAC CTT TGC CCA AGT CAT CT |
| GPx | CCA AGC TCA TCA CCT GGT CT | TCG ATG TCA ATG GTC TGG AA |
| GSR | ACT TGC CCA TCG ACT TTT TG | GGT GGC TGA AGA CCA CAG TT |
| CAT | CTG GAG CAC AGC ATC AAA TA | TCA TTC AGC ACG TTC ACA TAG A |
| HPRT | GAA CGT CTT GCT CGA GAT GTG A | TCC AGC AGG TCA GCA AAG AAT |

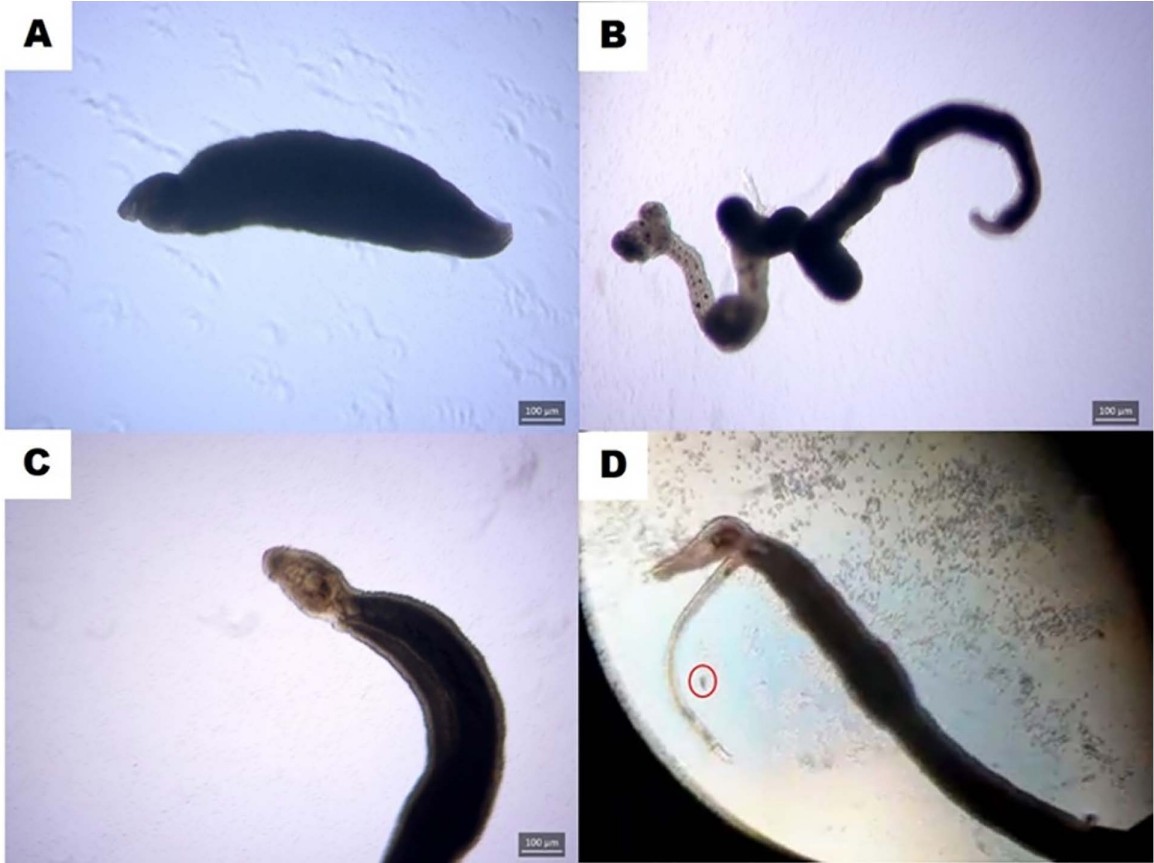

**Fig 1. Representative images illustrating the viability scoring system for adult *Schistosoma mansoni* worms cultured in vitro. (A)** Score 0 – Dead worm with fully darkened tegument and no movement. **(B)** Score 1 – Worm with darkened tegument, reduced motility, and impaired suckers. **(C)** Score 2 – Worm with partially damaged tegument and moderate motility. **(D)** Score 3 – Fully viable worm with intact tegument and active movement. The red circle indicates egg deposition, also used as a marker of viability. Image created by the authors.

## Dapsone hydroxylamine assay

Dapsone hydroxylamine (Santa Cruz Biotechnology, TX, USA) was tested in three different in vitro experimental models. First, assays were performed using only varying concentrations of dapsone hydroxylamine (12.5 µM, 25 µM, 50 µM, 100 µM, and 200 µM) to assess its direct effect on adult worms. The second model aimed to evaluate the ability of dapsone hydroxylamine to inhibit catalase activity *in vitro* in the presence of a pro-oxidant environment. For this, *S. mansoni* worm pairs were cultured in RPMI medium containing 400 µM of $H_2O_2$, a dose previously shown to cause ~50% mortality, along with 400 units of catalase and the different concentrations of dapsone hydroxylamine. A third model was tested by co-culturing HUVECs with worm pairs, 400 µM $H_2O_2$, and the various concentrations of dapsone hydroxylamine. All conditions were incubated at 37 °C with 5% $CO_2$ for 24 hours. Following incubation, worm viability was assessed using the same criteria as in the oxidative stress assays.

## Statistical analysis

For the PCR assays, gene expression differences between groups were analyzed using one-way ANOVA followed by Dunn's post hoc test. Outliers identified by mathematical criteria were excluded. Results are expressed as median and

interquartile range (IQR) or mean and standard deviation (SD). For the analysis of adult *S. mansoni* worm survival, group comparisons were performed using Fisher's exact test, and survival curves were compared using the Log-rank (Mantel-Cox) test. Statistical significance was defined as $p < 0.05$. All analyses were conducted using GraphPad Prism version 8.0 (GraphPad Software, San Diego, CA, USA).

## Results

### Expression of antioxidant genes in HUVECs

We initially assessed the expression of five antioxidant response-related genes in HUVECs co-cultured with adult *S. mansoni* worm pairs for 1, 3, and 6 hours. As shown in Fig 2 and S1 Table, the relative mRNA expression of NRF2 remained stable across all time points, both in HUVECs exposed to adult worms and in the LPS-stimulated control group. Similarly, SOD1 expression did not vary under any of the experimental conditions. The expression of GPX also remained unchanged over time in response to worm exposure. In contrast, GSR expression significantly decreased at 3 hours [0.883 (0.335)] compared to 1 hour [1.181 (0.539); $p < 0.05$], followed by a significant increase at 6 hours [1.268 (0.382); $p < 0.05$]. Notably, CAT mRNA expression progressively increased in HUVECs exposed to adult *S. mansoni*, with a significant difference between 1 hour [0.925 (0.590)] and 6 hours [1.547 (0.718); $p < 0.05$].

### Survival of adult *S. mansoni* worms after exposure to oxidative stress

The survival capacity of adult *S. mansoni* worms under pro-oxidant conditions was evaluated by culturing worm pairs in increasing concentrations of hydrogen peroxide ($H_2O_2$), either in the presence or absence of HUVECs. The distribution of viability scores differed among the cultures with increasing $H_2O_2$ concentrations. Worm death (score 0) was only observed at $H_2O_2$ concentrations starting from 400 µM. The frequency of score 0 increased progressively at higher $H_2O_2$ concentrations: 56.3% at 400 µM, 75% at 800 µM, and 100% worm death at 1600 µM $H_2O_2$ (Fig 3 and S2 Table).

In contrast, co-culturing worms with HUVECs under the same oxidative conditions yielded different survival outcomes. Viability score 3, indicating full tegument integrity and motility, and thus optimal viability, was observed even at higher $H_2O_2$ concentrations. Specifically, 50% of worms scored 3 at 400 µM, 33.3% at 800 µM, and 22.2% at 1600 µM $H_2O_2$. Notably, no worm deaths (score 0) were recorded at 800 µM in the presence of HUVECs, while 61.1% of worms were dead at 1600 µM (Fig 3 and S2 Table).

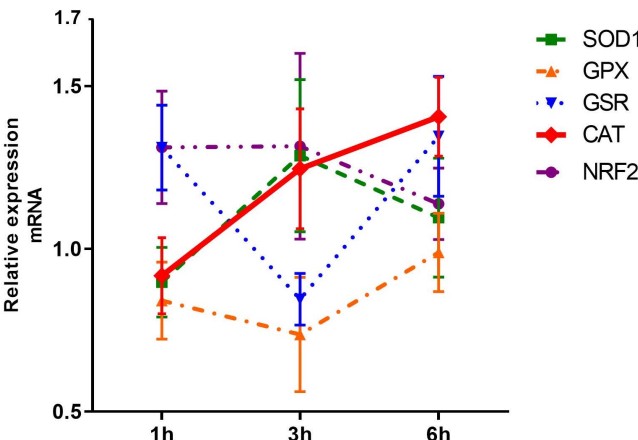

**Fig 2. Temporal expression of antioxidant genes in HUVECs exposed to adult *Schistosoma mansoni*.** Relative mRNA expression of antioxidant genes (SOD1, GPX, GSR, CAT, and NRF2) in human umbilical vein endothelial cells (HUVECs) after co-culture with adult *S. mansoni* worm pairs for 1, 3, and 6 hours.

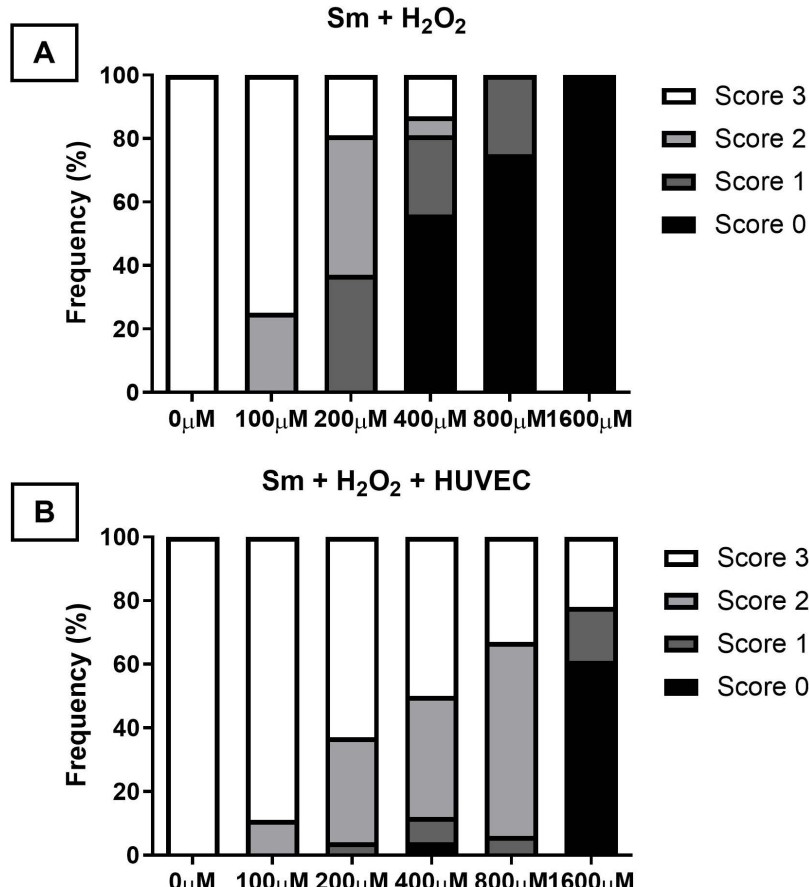

**Fig 3. Distribution of *S. mansoni* worm viability scores after exposure to oxidative stress with and without HUVECs.** Viability scores of adult *Schistosoma mansoni* worms after 24-hour exposure to increasing concentrations of hydrogen peroxide ($H_2O_2$), in the absence **(A)** or presence **(B)** of HUVECs. Worm viability was categorized into four scores: score 3 (intact tegument and active motility), score 2 (partial tegument damage and moderate motility), score 1 (darkened tegument and low motility), and score 0 (complete loss of motility and tegument integrity, indicating death). In the presence of HUVECs, a higher proportion of worms maintained viability (score 3), even at elevated $H_2O_2$ concentrations, with no deaths observed at 800 µM. In contrast, exposure to $H_2O_2$ alone resulted in a dose-dependent decline in viability, with complete mortality at 1600 µM.

To compare survival outcomes across experimental conditions, worms were categorized into two groups based on viability scores: high viability group (HVG; scores 2 and 3) and low viability group (LVG; scores 0 and 1). A significant increase in worm survival was observed at $H_2O_2$ concentrations of 400 µM and 800 µM when worms were co-cultured with HUVECs, compared to those exposed to $H_2O_2$ alone (Fig 4A and S3 Table).

The survival curves of adult *S. mansoni* worms exposed to oxidative stress were analyzed under three different experimental conditions to determine whether the addition of catalase mimics the protective effect observed in co-culture with HUVECs. In the absence of HUVECs or catalase, worm survival declined markedly from 400 µM $H_2O_2$ onward. In contrast, co-cultures with HUVECs showed mortality only at the highest concentration tested (1,600 µM $H_2O_2$). No statistically significant differences were observed between the survival curves of worms cultured with HUVECs and those treated with catalase, suggesting that HUVEC-mediated protection may be attributed, at least in part, to the secretion of antioxidant enzymes such as catalase (Fig 4B and S3 Table).

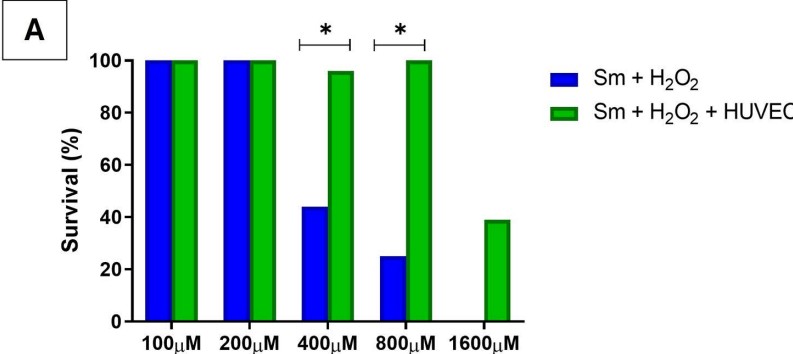

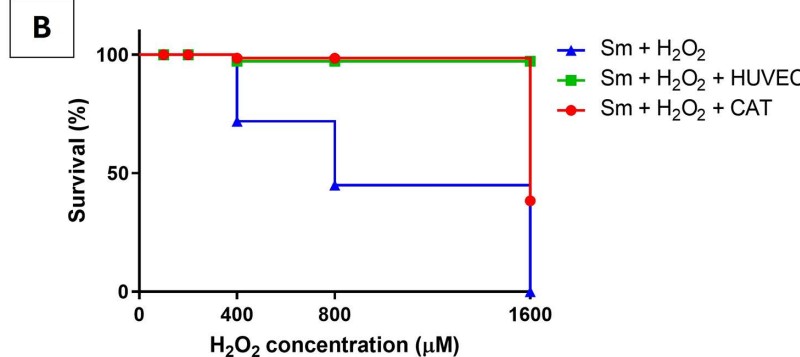

**Fig 4. Protective effect of HUVECs on *S. mansoni* worm survival under oxidative stress. (A)** Percentage of adult *Schistosoma mansoni* worms classified as high viability (scores 2–3) after exposure to increasing concentrations of hydrogen peroxide ($H_2O_2$), either in the absence (Sm + $H_2O_2$) or presence (Sm + $H_2O_2$ + HUVEC) of human umbilical vein endothelial cells (HUVECs). Co-culture with HUVECs significantly improved worm survival at 200, 400, and 800 μM $H_2O_2$, compared to exposure to $H_2O_2$ alone ($p < 0.001$, Fisher's exact test). **(B)** Survival analysis of adult Schistosoma mansoni worms exposed to increasing concentrations of hydrogen peroxide ($H_2O_2$) in three experimental models: worms incubated with $H_2O_2$ alone (blue line), co-cultured with HUVECs (green line), or treated with exogenous catalase (CAT, red line). Worms cultured without HUVECs or catalase exhibited reduced survival starting at 400 μM $H_2O_2$. In contrast, both co-culture with HUVECs and catalase treatment preserved worm viability, with mortality observed only at the highest concentration (1600 μM). No significant difference was detected between the HUVEC and catalase models, suggesting that the protective effect of endothelial cells may be mediated, at least in part, by antioxidant enzymes such as catalase.

## Dapsone reversed the in vitro effect of catalase

Considering the ability of hydroxylamine to inhibit catalase, the objective of this study was to evaluate the potential of dapsone's active metabolite, dapsone hydroxylamine (DDS-NOH), to block catalase activity and thus promote parasite death. To assess the direct effect of DDS-NOH on adult worms, the parasites were cultured in the absence of catalase, $H_2O_2$, or HUVECs and exposed them to different concentrations of DDS-NOH. A total of 100% of worms showed low viability at 100 μM, and 83.3% at 200 μM. At 50 μM, only 38.8% of the worms showed low viability, while the lowest concentrations, 25 μM and 12.5 μM, resulted in 5.5% of worms with low viability (Table 2).

High mortality in worms cultured with 100 and 200 μM DDS-NOH suggests that death was caused by a direct effect of the drug. To assess the ability of DDS-NOH to inhibit catalase, lower concentrations were used; therefore, the following results were obtained using only 50 μM DDS-NOH. When adult worms were cultured with $H_2O_2$, an increased percentage of worms with low viability (81%) was observed. This frequency dropped considerably (to 30%) when catalase was added to the culture. However, when worms were cultured with $H_2O_2$, catalase, and 50 μM DDS-NOH, the percentage of worms

**Table 2. Percentage of worm survival after exposure to different concentrations of Dapsone hydroxylamine.**

| Drug | Concentration in µM | Total Number of worms tested | Death Rate (%) |
|---|---|---|---|
| **No drug** | – | 18 | 0 |
| **Dapsone hydroxylamine** | 12.5 | 18 | 5.5 |
| | 25 | 18 | 5.5 |
| | 50 | 18 | 38.8 |
| | 100 | 6 | 100 |
| | 200 | 6 | 83.3 |

The percentage of worm mortality was determined after exposure to each concentration. Control groups were maintained without drug exposure. Data are presented as the number of worms tested and corresponding mortality rates for each condition.

with low viability increased again, suggesting that DDS-NOH was able to inhibit catalase activity in vitro. Interestingly, when worms were cultured with $H_2O_2$, catalase, and DDS-NOH in the presence of HUVECs, no loss in worm viability was observed. This finding suggests that, beyond catalase, HUVECs may protect adult *S. mansoni* worms through additional mechanisms (Fig 5 and S4 Table).

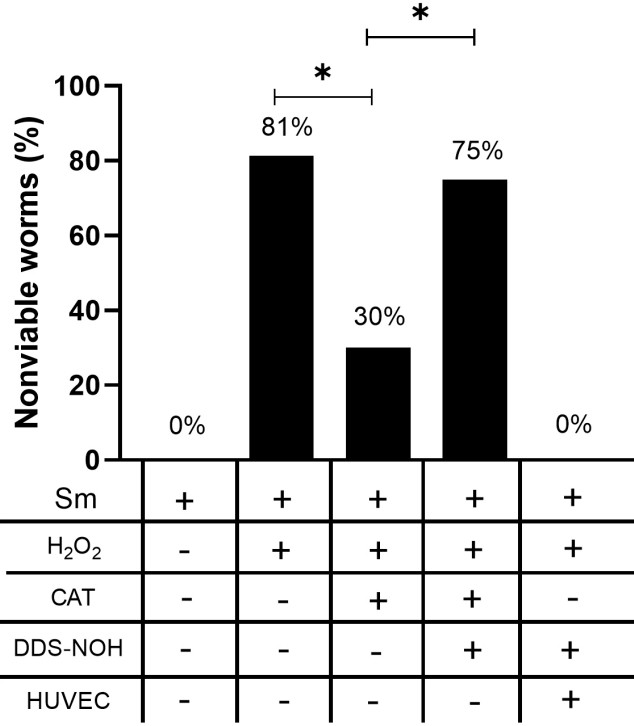

**Fig 5. Effect of DDS-NOH on the viability of adult *S. mansoni* worms under oxidative stress in different experimental conditions.** The percentage of nonviable worms was assessed following exposure to $H_2O_2$ (400 µM), with or without the addition of catalase (CAT), DDS-NOH (50 µM), and/or HUVECs. Catalase reduced the mortality induced by $H_2O_2$, an effect that was reversed in the presence of DDS-NOH, suggesting catalase inhibition. Interestingly, co-culture with HUVECs preserved worm viability even in the presence of DDS-NOH, indicating that HUVECs may provide additional protective mechanisms beyond catalase activity. Bars represent the percentage of nonviable worms in each experimental condition; horizontal lines indicate statistically significant differences ($p < 0.05$, Fisher's exact test).

## Discussion

The findings of this study reveal a previously unrecognized role of endothelial cells in promoting the survival of *S. mansoni* under oxidative stress, adding a novel layer to the understanding of host-parasite interactions. While parasite survival in the vascular system has traditionally been explained by immune evasion strategies—such as the acquisition of host antigens [18], molecular mimicry [19], tegumental resistance [20], and immunomodulatory proteins and extracellular vesicles [21–23], the current results suggest that host vascular endothelial cells may also directly contribute to parasite protection.

The most striking and original result is the observation that adult *S. mansoni* worms maintained high viability when co-cultured with human umbilical vein endothelial cells (HUVECs), even in the presence of $H_2O_2$, a potent pro-oxidant. This protective effect appears to be at least partially mediated by catalase, an antioxidant enzyme produced by HUVECs [24]. The enhancement of catalase expression in co-culture conditions, accompanied by higher worm viability, supports the hypothesis that endothelial antioxidant mechanisms play a direct role in parasite resilience.

These findings align with previous studies indicating the importance of oxidative stress resistance in *S. mansoni* survival [25]. However, the demonstration that host-derived antioxidant responses can compensate for exogenous oxidative pressure expands the current understanding of host support in chronic infections. Importantly, this study demonstrates that dapsone hydroxylamine (DDS-NOH) significantly reduces worm viability under oxidative conditions. DDS-NOH has been reported to inhibit catalase activity in mammalian cells [26,27], suggesting that this mechanism may also contribute to its effect on *S. mansoni*. However, since catalase activity was not directly assessed in our experimental model, further studies are needed to confirm this mechanism in the context of schistosomiasis. Nonetheless, these findings support the idea that targeting antioxidant pathways may represent a promising strategy to sensitize adult worms to oxidative damage.

An unexpected yet revealing observation was that worm viability remained preserved in the presence of HUVECs, even under conditions of pharmacological catalase inhibition. This finding suggests that endothelial cells may contribute additional cytoprotective mechanisms beyond catalase activity, including the secretion of soluble mediators, modulation of the redox microenvironment, or provision of metabolic substrates. Endothelial cells are known to release various antioxidant molecules, such as glutathione, nitric oxide, and prostaglandins, which can mitigate oxidative stress and modulate inflammatory responses [28–30]. These findings challenge the notion that antioxidant enzyme activity alone is sufficient to explain host-mediated protection and point to the complexity of host-parasite biochemical crosstalk.

Despite the robustness of the in vitro model employed, certain limitations must be acknowledged. The study was conducted under controlled laboratory conditions that may not fully replicate the dynamics of in vivo parasite-endothelial interactions, such as immune cell recruitment, blood flow, and tissue architecture. Additionally, the precise molecular pathways underlying HUVEC-mediated protection were not identified and require further exploration.

The implications of these findings are twofold. First, they highlight the importance of considering host tissue contributions beyond immune evasion or parasite-intrinsic adaptations when investigating the persistence of *S. mansoni* in the vasculature. Second, they identify endothelial cells as potential facilitators of parasite survival, actively shaping the microenvironment in a way that favors worm maintenance. This discovery opens new perspectives on the vascular niche as a supportive habitat for adult worms, which may be critical for their long-term survival despite the presence of immune effector cells such as eosinophils and other producers of reactive oxygen species.

Rather than being passive barriers, endothelial cells emerge here as active modulators of parasite viability, with potential implications for both pathogenesis and treatment. Understanding the molecular mechanisms involved in this protection—particularly the identity of soluble mediators and redox-modifying factors—may uncover novel therapeutic targets and deepen our knowledge of host–parasite interactions at the vascular interface.

## Conclusion

This study reveals a novel role of human endothelial cells in promoting *Schistosoma mansoni* viability under oxidative stress. Our findings suggest that endothelial cells, which remain in constant contact with adult worms in the vasculature,

contribute actively to parasite maintenance. While catalase activity appears to participate in this protective effect, the persistence of worm viability despite catalase inhibition points to additional, yet unidentified, protective pathways. Altogether, these results challenge the traditional view of immune evasion as the sole mechanism of worm persistence and highlight the vascular endothelium as a critical, and potentially targetable, player in schistosomiasis pathophysiology.

## Supporting information

**S1 Table. Relative expression levels of antioxidant response genes in HUVECs.** Relative expression values of NRF2, SOD1, GPx, GSR, and CAT genes. Expression levels were normalized to the endogenous control gene HPRT. Relative expression (ddCt) values were calculated using the median of untreated controls as the calibrator sample.
(XLSX)

**S2 Table. Viability of adult *Schistosoma mansoni* worms after exposure to hydrogen peroxide.** (A) Number of adult worms after 24-hour exposure to increasing concentrations of hydrogen peroxide ($H_2O_2$). (B) Number of adult worms after 24-hour exposure to increasing concentrations of $H_2O_2$ in the presence of HUVECs. Worm viability was categorized into four scores: score 3 (intact tegument and active motility), score 2 (partial tegument damage and moderate motility), score 1 (darkened tegument and low motility), and score 0 (complete loss of motility and tegument integrity, indicating death).
(XLSX)

**S3 Table. Number of *Schistosoma mansoni* adult worms with high or low viability after 24-hour exposure to increasing concentrations of hydrogen peroxide ($H_2O_2$).** Worm viability was assessed under three experimental conditions: (A) exposure to $H_2O_2$ alone (100–1600 µM), (B) exposure to $H_2O_2$ in the presence of HUVECs, and (C) exposure to $H_2O_2$ with catalase supplementation.
(XLSX)

**S4 Table. Number of *Schistosoma mansoni* adult worms with high or low viability after exposure to different conditions.** Worms were exposed to $H_2O_2$ alone, $H_2O_2$ with catalase, or $H_2O_2$ with catalase plus 50 µM DDS-NOH, and worms exposed to $H_2O_2$ in the presence of HUVECs and DDS-NOH.
(XLSX)

## Acknowledgments

We would like to thank Dr. Ana Moretti, Ph.D. and Dr. Heraldo Possolo de Souza, M.D., from the São Paulo State University Medical School (FMUSP-Brazil), for generously donating the HUVEC cells used in this study.

## Author contributions

**Conceptualization:** Bruna Oliveira Lopes Souza, Sânzio Silva Santana, Karine Araujo Damasceno, Isadora Cristina de Siqueira, Marilda de Souza Gonçalves, Dalila Luciola Zanette, Thassila Nogueira Pitanga, Ricardo Riccio Oliveira.

**Data curation:** Bruna Oliveira Lopes Souza, Ricardo Riccio Oliveira.

**Formal analysis:** Bruna Oliveira Lopes Souza, Dalila Luciola Zanette, Ricardo Riccio Oliveira.

**Funding acquisition:** Dalila Luciola Zanette, Thassila Nogueira Pitanga, Ricardo Riccio Oliveira.

**Investigation:** Bruna Oliveira Lopes Souza, Ronald Alves dos Santos, Kelvin Edson Marques de Jesus, Juliana Bezerra Dória Lima, Andressa Moreira Lima, Brenda Rodrigues Brito Cunha Silva, Fernanda Freitas Costa, Lorena Conceição Queiroz, Sânzio Silva Santana, Dalila Luciola Zanette, Ricardo Riccio Oliveira.

**Methodology:** Bruna Oliveira Lopes Souza, Ronald Alves dos Santos, Kelvin Edson Marques de Jesus, Sânzio Silva Santana, Karine Araujo Damasceno, Dalila Luciola Zanette, Thassila Nogueira Pitanga, Ricardo Riccio Oliveira.

**Project administration:** Bruna Oliveira Lopes Souza, Ricardo Riccio Oliveira.

**Resources:** Bruna Oliveira Lopes Souza, Marilda de Souza Gonçalves, Dalila Luciola Zanette, Ricardo Riccio Oliveira.

**Software:** Bruna Oliveira Lopes Souza, Ricardo Riccio Oliveira.

**Supervision:** Ricardo Riccio Oliveira.

**Validation:** Bruna Oliveira Lopes Souza, Ricardo Riccio Oliveira.

**Visualization:** Bruna Oliveira Lopes Souza, Ricardo Riccio Oliveira.

**Writing – original draft:** Bruna Oliveira Lopes Souza, Camilla Almeida Menezes, Ricardo Riccio Oliveira.

**Writing – review & editing:** Bruna Oliveira Lopes Souza, Camilla Almeida Menezes, Karine Araujo Damasceno, Isadora Cristina de Siqueira, Marilda de Souza Gonçalves, Dalila Luciola Zanette, Thassila Nogueira Pitanga, Ricardo Riccio Oliveira.

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
