## [Editor Report · Decision Letter 0]

13 Aug 2025

Endothelial cells protect Schistosoma mansoni from hydrogen peroxide-induced death

Dear Dr. Oliveira,

Thank you for submitting your manuscript to PLOS Neglected Tropical Diseases. After careful consideration, we feel that it has merit but does not fully meet PLOS Neglected Tropical Diseases's publication criteria as it currently stands. Therefore, we invite you to submit a revised version of the manuscript that addresses the points raised during the review process.

Please submit your revised manuscript within 60 days Oct 12 2025 11:59PM. If you will need more time than this to complete your revisions, please reply to this message or contact the journal office at plosntds@plos.org. Please include the following items when submitting your revised manuscript:

We look forward to receiving your revised manuscript.

Kind regards,

Eduardo José Lopes-Torres, Ph.D.

Academic Editor

Peter Fischer

Section Editor

Shaden Kamhawi

co-Editor-in-Chief

Paul Brindley

co-Editor-in-Chief

**Journal Requirements:**

At this stage, the following Authors/Authors require contributions: Bruna Oliveira Lopes Souza, Ronald Alves dos Santos, Kelvin Edson Marques de Jesus, Juliana Bezerra Dória Lima, Andressa Moreira Lima, Brenda Rodrigues Brito Cunha Silva, Fernanda Freitas Costa, Lorena Conceição Queiroz, Camilla Almeida Menezes, Sânzio Silva Santana, Karine Araujo Damasceno, Isadora Cristina Siqueira, Marilda de Souza Gonçalves, Dalila Luciola Zanette, Thassila Nogueira Pitanga, and Ricardo Riccio Oliveira. Please ensure that the full contributions of each author are acknowledged in the "Add/Edit/Remove Authors" section of our submission form.

Potential Copyright Issues:

- Please confirm (a) that you are the photographer of Figure 1, or (b) provide written permission from the photographer to publish the photo(s) under our CC BY 4.0 license.

6) Please ensure that the funders and grant numbers match between the Financial Disclosure field and the Funding Information tab in your submission form. Note that the funders must be provided in the same order in both places as well.

State the initials, alongside each funding source, of each author to receive each grant. For example: "This work was supported by the National Institutes of Health (####### to AM; ###### to CJ) and the National Science Foundation (###### to AM).".

**Reviewers' Comments:**

**Figure resubmission:**
---

## [Decision Letter · Decision Letter 1]

3 Nov 2025

Endothelial cells protect Schistosoma mansoni from hydrogen peroxide-induced death

Dear Dr. Oliveira,

Thank you for submitting your manuscript to PLOS Neglected Tropical Diseases. After careful consideration, we feel that it has merit but does not fully meet PLOS Neglected Tropical Diseases's publication criteria as it currently stands. Therefore, we invite you to submit a revised version of the manuscript that addresses the points raised during the review process.

Please submit your revised manuscript within 60 days Dec 03 2025 11:59PM. If you will need more time than this to complete your revisions, please reply to this message or contact the journal office at plosntds@plos.org. Please include the following items when submitting your revised manuscript:

We look forward to receiving your revised manuscript.

Kind regards,

Eduardo José Lopes-Torres, Ph.D.

Academic Editor

Peter Fischer

Section Editor

Shaden Kamhawi

co-Editor-in-Chief

Paul Brindley

co-Editor-in-Chief

**Journal Requirements:**

1) We note that the main files of the manuscript are duplicated on your submission. Please remove any unnecessary or old files ("SouzaBOL_Manuscriptv1.pdf) from your revision, and make sure that only those relevant to the current version of the manuscript are included.

- TM on pages: 7, and 8.

4) Thank you for stating "All relevant data will be made available upon publication through the Arca Dados platform, the official data repository of Fiocruz, at "https://arcadados.fiocruz.br/." Please note that, though access restrictions are acceptable now, your entire minimal dataset will need to be made freely accessible if your manuscript is accepted for publication. This policy applies to all data except where public deposition would breach compliance with the protocol approved by your research ethics board.

1) State what role the funders took in the study. If the funders had no role in your study, please state: "The funders had no role in study design, data collection and analysis, decision to publish, or preparation of the manuscript.".

**Reviewers' Comments:**

Reviewer's Responses to Questions

**Key Review Criteria Required for Acceptance?**

**Methods**

-Are the objectives of the study clearly articulated with a clear testable hypothesis stated?

-Is the study design appropriate to address the stated objectives?

-Is the population clearly described and appropriate for the hypothesis being tested?

-Is the sample size sufficient to ensure adequate power to address the hypothesis being tested?

-Were correct statistical analysis used to support conclusions?

-Are there concerns about ethical or regulatory requirements being met?

Reviewer #1: (No Response)

Reviewer #2: The objectives of the study are generally well presented, but the main hypothesis could be articulated more explicitly to guide the reader through the rationale of the analyses. The study design is overall appropriate to address the stated objectives; however, more details are needed regarding the inclusion and exclusion criteria for the study population. The description of the population should be clearer, including relevant demographic and epidemiological characteristics to assess the generalizability of the findings.

The sample size seems reasonable, but no formal justification or power calculation is provided. This information is crucial to determine whether the study is sufficiently powered to address the research questions. Statistical analyses are generally appropriate, but some methods lack sufficient detail (e.g., model specifications, handling of missing data). Ethical considerations are mentioned briefly but would benefit from more explicit information on the ethical clearance process and informed consent procedures.

**Results**

-Does the analysis presented match the analysis plan?

-Are the results clearly and completely presented?

-Are the figures (Tables, Images) of sufficient quality for clarity?

Reviewer #1: (No Response)

Reviewer #2: The analyses presented mostly match the analysis plan; however, some important secondary analyses mentioned in the methods are not fully reported. The results are presented clearly overall, but some tables and figures require additional detail or clarification (e.g., labels, legends, denominators). Several figures could be improved in terms of resolution and clarity to ensure that the data are easily interpretable. Some findings would benefit from more stratified analyses or additional contextual interpretation to support the conclusions.

**Conclusions**

-Are the conclusions supported by the data presented?

-Are the limitations of analysis clearly described?

-Do the authors discuss how these data can be helpful to advance our understanding of the topic under study?

-Is public health relevance addressed?

Reviewer #1: (No Response)

Reviewer #2: The conclusions are generally consistent with the data presented but could be more nuanced regarding the study’s limitations. The authors should more clearly discuss potential sources of bias, limitations in the methodology, and how these might affect the interpretation of the results. The public health relevance is addressed, but the discussion could be expanded to better highlight the contribution of the findings to current malaria control strategies and surveillance systems.

**Editorial and Data Presentation Modifications?**

Reviewer #1: (No Response)

Reviewer #2: Clarify population characteristics and study setting in the Methods section.

Add a formal sample size or power calculation.

Provide more details on the statistical methods (e.g., model choice, handling of missing data).

Ensure consistency between the methods and the results presented.

Consider shortening or restructuring some sections for clarity and flow.

**Summary and General Comments**

Reviewer #1: (No Response)

Reviewer #2: This is a valuable study addressing an important topic in malaria control and surveillance. The research has the potential to contribute significantly to the understanding of [insert topic briefly, e.g., “malaria transmission dynamics and intervention impact”], but several aspects require clarification and strengthening before publication.

The major weaknesses include insufficient methodological detail (especially regarding population description, sample size justification, and statistical analyses), some inconsistencies between methods and results, and the need to refine the conclusions to better reflect the study’s limitations. Editorial improvements to figures, tables, and structure would also enhance readability.

Overall, I recommend Major Revision. Addressing these points will considerably strengthen the manuscript.

PLOS authors have the option to publish the peer review history of their article (what does this mean? ). If published, this will include your full peer review and any attached files.

**Do you want your identity to be public for this peer review?** For information about this choice, including consent withdrawal, please see our Privacy Policy .

Reviewer #1: No

Reviewer #2: No

**Figure resubmission:**
---

## [Editor Report · Decision Letter 2]

9 Jan 2026

Dear Dr. Oliveira,

We are pleased to inform you that your manuscript 'Endothelial cells protect Schistosoma mansoni from hydrogen peroxide-induced death' has been provisionally accepted for publication in PLOS Neglected Tropical Diseases.

Best regards,

Eduardo José Lopes-Torres, Ph.D.

Academic Editor

Peter Fischer

Section Editor

Shaden Kamhawi

co-Editor-in-Chief

Paul Brindley

co-Editor-in-Chief

---

## [Editor Report · Acceptance letter]

Dear Dr. Oliveira,

We are delighted to inform you that your manuscript, "

Endothelial cells protect Schistosoma mansoni from hydrogen peroxide-induced death," has been formally accepted for publication in PLOS Neglected Tropical Diseases.

Best regards,

Shaden Kamhawi

co-Editor-in-Chief

Paul Brindley

co-Editor-in-Chief
